# OpenReview forum: "Controllable Text Generation in the Instruction-Tuning Era"
_TMLR — Rejected by TMLR_

### Review · Reviewer_KtdZ · 2024-04-25

**Summary Of Contributions:**

1. The research highlights the evolution of controllable text generation, emphasizing the shift from traditional methods to instruction-tuning and prompting-based approaches. These newer methods have shown promise in achieving better control over text outputs, particularly in adhering to specific stylistic and structural constraints.
2. A significant contribution of this work is the development of ConGenBench, a comprehensive benchmark comprising 17 different controllable generation tasks. This testbed is used to evaluate various methods on instruction-tuned LLMs, revealing that prompting-based approaches often surpass traditional controllable generation techniques in performance.
3. The study details an innovative algorithm that enables the creation of constraint-specific classification datasets directly from intended prompt datasets. This approach uses in-context capabilities of LLMs to generate and classify text, facilitating the study of more nuanced constraints without the need for extensive dataset collection.
4. Experimental results indicate that while prompting-based methods excel in stylistic tasks, they tend to lag in structural tasks, highlighting a gap that needs addressing in future research.
5. The paper also discusses the limitations of existing datasets used for training constraint classifiers, such as their inflexibility and the potential misalignment with the deployment environment's actual needs.
6. Lastly, the research calls for further exploration into more diverse and challenging constraints to advance the field of controllable text generation, suggesting that future work should also consider the integration of more complex and varied constraints.

**Audience:**

Yes

**Broader Impact Concerns:**

Incorporating more popular LLMs into the study could significantly enhance its relevance and applicability. This inclusion would likely attract broader attention and impact, as these models are at the forefront of current research and application. By addressing this gap, the paper could position itself as a pivotal resource in the ongoing development and evaluation of controllable text generation technologies.

**Claims And Evidence:**

Yes

**Requested Changes:**

The summary of contributions needs to be expanded to provide more detailed explanations of the key findings and technological advancements. The current summary, presented in three bullet points, is too concise and does not adequately reflect the depth and implications of the research. A more elaborate discussion would better showcase the significance of the benchmark and its novel contributions to the field.

**Strengths And Weaknesses:**

# Strengths:

1. The paper is well-written, offering clarity and precision in its descriptions and arguments, making it accessible and informative for readers with varied expertise in the field of natural language processing.
2. The authors have provided a thorough discussion on the considerations and challenges in building the ConGenBench. This benchmark is meticulously detailed, explaining the criteria and methodologies used, which highlights the robustness and depth of the research.
3. Figure 3 stands out for its clarity and completeness. It successfully illustrates the comprehensive aspects included in the benchmark, serving as an effective visual aid that enhances understanding of the complex data and scenarios covered.

# Weaknesses:

1. Table 1 lacks crucial information about the model sizes. This omission makes it difficult for readers to accurately compare the performance and efficiency of different models presented, which is critical for evaluating their practical applicability.
2. Additionally, the table does not cover many popular LLMs like Vicuna, Alpaca, GPT, and Llama. Including these models is essential for a comprehensive comparative analysis, as these models represent significant advancements and are widely used in the community.

---

> ### Author Response · Authors · 2024-05-27
> **Writing and Application to Largest Available Language Models**
>
> We appreciate the reviewer's feedback on the need to expand and provide more detailed explanations of the findings of the paper. Before reading this response we would like to direct the reviewer to the general official comment we have posted. That comment addresses the changes made and common concerns that multiple reviewers shared. In this response, we focus on the individual concerns of the reviewer.
>
> We have attempted to revise the writing to better reflect the implications of the work both in the introduction and contribution bullet point list. The reviewer also asked for clarification regarding the model sizes. All models tested are around 7 billion parameters in scale, the main paper and appendix have been rewritten to explicitly mention this fact.
> We share the reviewer's enthusiasm concerning how this same study could have greater relevance if it was conducted on the most popular, powerful, commercial language models like GPT4, Claude, Gemini etc, however, these are unfortunately not accessible at the moment and so we are limited to studying models in the 7 billion parameter scale.

---

### Review · Reviewer_4FxL · 2024-05-04

**Summary Of Contributions:**

The authors provide a novel method to evaluate the performance of
instruction-tuned language models with controlled generation.
This method utilizes regular evaluation datasets that lack
constraint classifiers, as it generates the constraints
automatically. With this method, the authors create a new evaluation
suite for controllable generation methods, ConGenBench, and
evaluate several state-of-the-art models and methods on it.
They find that in some tasks, the models outperform humans,
and that for instruction-tuned models, zero-shot and few-shot
prompting outperform the leading controllable generation methods.

**Audience:**

Yes

**Claims And Evidence:**

No

**Requested Changes:**

- I would like to see the individual standard deviations for each method and model in all tables, instead
  of having a standard deviation averaged across all methods and models for each task.
  This will provide more insights into the stability of the results and if the results
  are statistically significant.
- Preferably, more samples should be drawn such that the standard deviation is lower.
  This may require significant effort, but would strengthen the results and make them more reliable.
- Evaluations with more instruction-tuned models on different tasks and methods
  should be conducted to strengthen the claims made in the paper.
  The set of instruction-tuned models used in Table 1 are reasonable (Mistral-7B Instruct, Falcon Instruct, MPT Instruct),
  and would be a good starting point for further evaluations.
- Several version of Mistral-7B Instruct have been released, to facilitate reproducibility, please provide the exact version used in the experiments.
- Table 2 is out of the margins, it should be adjusted so that it fits within the margins.
- "On the topic control task(Table 2, the ZeroShot Prompting approach outperforms the human baseline by"
  Please fix the reference to Table 2.

**Strengths And Weaknesses:**

## Strengths

- The problem of evaluating these methods on larger and more diverse
  datasets is important, and the authors provide a method to expand
  regular evaluation datasets to evaluate controlled generation
  without needing human annotations.
- The datasets the authors compile are large, diverse, and cover a
  wide range of tasks. This is a significant contribution to the field.
- The results are insightful and can help guide future research in
  controllable generation.

## Weaknesses
- Generally, the average standard deviation across different results is quite high,
  suggesting that the results are not very stable.
  For instance, in Table 3, for the Fluency task, the average standard deviation is 0.31,
  with scores ranging from 6.98 to 7.2, showing that the results may not be statistically significant.
  Singular standard deviations for each method and model would provide more insights into the stability of the results.
- The authors claim to provide an analysis for instruction-tuned models,
  but the only model they thoroughly evaluate on all tasks is Mistral-7B Instruct.
  Only in Table 1, they evaluate other models, however they still only evaluate
  Mistral-7B Instruct on methods other than zero-shot and few-shot prompting.

---

> ### Author Response · Authors · 2024-05-27
> **Standard Deviation and Statistical Significance**
>
> We extend our gratitude for the reviewers keen eyes in spotting writing and formatting mistakes, we have made the required changes. Before reading this response we would like to direct the reviewer to the general official comment we have posted. That comment addresses the changes made and common concerns that multiple reviewers shared. In this response we focus on the individual concerns of the reviewer.
>
> On the reviewers request, the tables in the paper have been updated to reflect the standard deviation between the 3 annotator scores for each task setting, model and method combination. We prefer to avoid interpreting this as a standard deviation measure that can be used to construct a confidence interval because we believe 3 samples is too few to faithfully declare statistical significance. We agree with the reviewer that taking more than 3 human evaluations per sample output would make the results more robust, however as explained in the general comment on 'Limited Scope of Experiments' the cost of adding additional survey participants was prohibitively high. If we compute the t-statistic and construct confidence intervals using this measure then the results are not significant at p<0.05 level. Most results are significant at p-levels around p<0.15 to p<0.2
> We share the reviewers' concerns that due to the high variance of results we should be careful when interpreting the results. This is why in our discussions of the results we focus only on the signals which are strong (margin is high relative to deviation which is related to high t-statistic value) and consistent (visible across almost all datasets and methods). We refrain from declaring any controllable text generation method better than another and refrain from declaring any model better than others.
>
> We do believe, however, that there is sufficient evidence to make the key claims forwarded by the paper -
> 1. Controllable text generations consistently perform better than baselines like greedy decoding and reranking
> 2. Controllable text generation methods consistently perform worse than prompt-based baselines on instruction-tuned LLMs
> 3. Prompt-based baselines perform competitively with human performance on most stylistic tasks, while structural constraint tasks remain challenging.

---

### Review · Reviewer_yukQ · 2024-05-19

**Summary Of Contributions:**

This paper compares the efficacy of controlled text generation techniques (based on some form of constrained decoding) with prompting for controlled text generation for instruction-tuned LLMs. The central contribution of the paper is that the finding that prompting techniques are better than specialized decoding techniques for instruction-tuned LLMs, and in some cases the LLMs get results comparable to human generations.

In addition, the paper also discusses a synthetic data generation algorithm to use an auxiliary LLM to label generations of the target LLM, and talk about creation of a new benchmark ConGenBench comprising controlled text generation tasks over 17 datasets.

**Audience:**

Yes

**Broader Impact Concerns:**

It is important to develop techniques to generate harmless and helpful text. While the paper shows that simple prompting can help with it, isn't it equally possible to flip those constraints and generate harmful content? The authors should consider adding an impact statement to address this use of their findings.

**Claims And Evidence:**

No

**Requested Changes:**

1. The evaluation should cover multiple instruction-tuned LLMs and apply the baselines and prompting methods uniformly so that more convincing conclusions can be drawn.
2. The paper should explain what is so novel about Algorithm 1 and what it is in the paper.
3. What are the new contributions in constructing ConGenBench compared to existing datasets?

**Strengths And Weaknesses:**

Strengths
1. Apparently the constrained decoding methods for controlled generation have not been compared against simple prompting for instruction-tuned LLMs.
2. The paper through human assessment provides evidence that the latter are better than the specialized methods.
3. It evaluates the constrained decoding and prompting methods on 7 datasets on the corresponding stylistic tasks and some structural constraints restricting the generation (e.g., restriction on the number of words).

Weaknesses
1. The claims of the paper are drawn primarily by evaluating a _single_ instruction-tuned LLM (MistralInstruct7B). The paper does not answer questions like whether the claims hold broadly, how the LLM size or training setups affect the results, etc.
2. For two tasks (Table 1), two additional LLMs are considers but the controlled text generation methods are not applied uniformly to them.
3. LLMs are used quite regularly for labeling and evaluation. The labeling algorithm (Algorithm 1) is not novel in that sense.
4. It is not even clear what role Algorithm 1 plays in this of the paper except for a mention in Sec 6 about how it can be used.
5. The paper is not clear about what is the value addition that it makes by proposing ConGenBench. It looks as if it is simple aggregation of existing datasets and an enumeration of some constraints.
6. The paper makes no promises about making their data and code available. If the benchmark is a claimed contribution, then I would expect these to be made available.

---

> ### Author Response · Authors · 2024-05-27
> **Novelty and Importance of Algorithm 1**
>
> We thank the reviewer for sharing their questions regarding the novelty of Algorithm 1 and the role it plays in the paper. The reviewer also brought up the fact that the model-method combinations are not uniform across all models and mentioned that we should add a section on the ethical impacts of controllable text generation (section added to appendix). Before reading this response we would like to direct the reviewer to the general official comment we have posted. That comment addresses the changes made and common concerns that multiple reviewers shared. In this response, we focus on the individual concerns of the reviewer.
>
> We hope the section on the Limited Scope of Experiments provides our justification for why we have not conducted as extensive an evaluation as possible. The first experiment on toxicity and sentiment shows that across the 3 Language Models studied, the results are quite similar across different methods. This suggests that the deeper explorations in the subsequent experiments on topic control, sensationalism, excitement and irony (on only Mistral7B) are a fair indication of how other similar models would perform.
>
>
> Algorithm 1, is used for the synthetic generation of the constraint dataset and hence constraint score function. It is true that the usage of Language Models to perform pseudo-labelling is not in itself novel, however, we feel that there are several reasons why the algorithm is both novel and an important contribution to the field.
>
> The algorithm is an important contribution to the field because of what it enables - the study of constraints that do not currently have a pre-curated constraint dataset associated with it. Prior work in controllable text generation is reliant on pre-curated datasets, severely limiting the scope and range of constraints that they study. For example, the algorithm is a vital component of this paper, without it we would not be able to study the constraints of sensationalism, excitement and irony as they do not have pre-curated datasets. With the introduction of this algorithm, researchers can now easily study diverse constraints like "disappointment, objectivity, matches the style of the Hobbit" etc.
>
>
> In terms of technical novelty, the algorithm is the first to provide a general-purpose pipeline that can convert any arbitrary prompt dataset into a synthetic constraint dataset that is required for the deployment of controllable text generation methods. The algorithm also involves a discussion that identifies a distribution/domain shift problem that is prevalent with existing constraint datasets and explains how the method circumvents this problem. This is not a general observation that is true in all cases where LLMs are used for pseudo labelling and is a unique technical insight relevant to controllable text generation and the constraint datasets that are typically used to construct score functions.

---

### Author Response · Authors · 2024-05-27
**General Comment to All Reviewers**

We thank all the reviewers for their insightful comments and feedback on the paper. After going through the feedback and understanding the desired changes we have updated the submission. The writing has been updated to address the reviewers' concerns and incorporate desired changes where it was most appropriate. We would first like to list the changes made, then respond to common concerns that multiple reviewers have shared or raised, and finally address individual reviewers' queries.

## Changes Made:
1. Experiment details made more clear (model sizes added to the main paper and Appendix, exact version specified in the appendix for reproducibility)
2. Standard deviations for individual surveys for each method and each metric added
3. Expanded discussion of contributions, emphasizing the importance of the results and the value of the ConGenBench testbed.
4. Miscellaneous formatting edits.

## Common Concerns:


### Limited Scope of Experiments:
All the reviewers rightly identify that this work only studies 3 instruction-tuned Language Models - Mistral, Falcon and MPT. Additionally, it is noted that Mistral is the only Language Model with deeper investigation, while the other Language Models do not have all methods run on their instruction-tuned variants and are not studied in every task setting. While we would have liked to study the most powerful open-sourced models (in the 40-70 billion parameter range) we were limited by our access to computational resources, and are unable to run controllable text generation methods on closed-sourced models like GPT4 or Claude because they do not allow us to prefix tune or access logits. One of the reviewers also points out that we have not performed ablations over model sizes, training setups etc. Given our computational resources, it is not feasible to retrain or perform ablations with the training setup for billion-parameter scale language models. While we can retrain models in the 100 million parameter scale, these models do not exhibit as much "emergent" in-context instruction following capabilities as the larger instruction-tuned LLMs and so were considered out of the scope of this paper.

We study around 8 different methods and 6 different tasks, this means adding even a single ablation or additional model would add 7200 HIITS on Amazon Mechanical Turk and raise the total cost significantly. We considered using automatic metrics instead of human evaluation, prior work sometimes trains an auxiliary classifier using an adjacent constraint dataset and uses the logits as a metric. However, as discussed in Section 4 of the paper we find there to be a significant distribution/domain shift between the task dataset and the constraint dataset. Additionally, our initial experiments suggested that automatic metrics had too high variance, and generally were not a reliable proxy.

In short, due to financial and computational constraints, we were faced with tradeoffs between exploring different models, methods and task settings.  We chose to prioritize a diversity of tasks because we felt it would give a richer understanding of how different methods performed under varying settings.

Continued Below.....

---

> ### Author Response · Authors · 2024-05-27
> **Continuation:**
>
> ### New contribution and value addition of ConGenBench:
> Multiple reviewers have asked for a more elaborate discussion on the significance of the benchmark and the value it adds. We have revised the writing to address this, however, the paper still does not have too detailed a discussion of the various intricacies of the ConGenBench testbed because we felt it more appropriate to state those clearly in the GitHub repository associated with the testbed. However, if the reviewers read this response and decide they think the following sections should also feature in the paper, we are more than happy to include them in the appendix.
>
> We commit to releasing our data and code as soon as the anonymity restrictions are raised. Overall the ConGenBench provides three key contributions:
> 1. Identification and compilation of several datasets that can be used for controllable text generation benchmarking in the future.
> The task datasets included are freely accessible generative task datasets from various other disciplines and problem settings of NLP. However, the majority of them have never been used to benchmark Controllable Text Generation methods before, and the identification of a diverse set of tasks that are both appropriate and interesting settings for controllable text generation is an important contribution of its own.
>
> 2. Standardization of problem setting and non-trivial task formulation for each dataset
> After identifying which datasets and tasks to include in the ConGenBench dataset, we then perform preprocessing specific to each dataset and convert each task into a standard format. This is because it is not obvious how some of these datasets should be used to evaluate controllable text generation, and there are several design decisions to be made when converting the datasets to a controllable text generation task. For example, we have included the Factual Error Correction task with the SciFact and FEVER datasets. These datasets come with a set of claims and evidence documents that support, refute or do not pertain to some claims in the datasets. In preprocessing, we selectively keep only the refuted claims, extract the most relevant evidence sentences and create a "controllable text generation" task which consists of a single prompt along with a task description. Another example is the CNNDailymail news continuation task, where different ways of cutting off the original article (e.g. using only the first 2 words vs using the first 3 sentences) can fundamentally change the nature of the task setting. Previous work has sometimes used the same datasets but with varying parameters for these design decisions, complicating efforts to compare different methods. ConGenBench not only establishes standardized parameter settings and dataset splits across all tasks, but it also makes the parameters plainly visible and easy to edit in the code. This enables future research to study different variations of each task with ease.
>
> 3. Ease of use, organization and flexibility
> ConGenBench is the first singular testbed that can claim to provide a more comprehensive evaluation suite for controllable text generation. We have worked to make the code easy to use, with minimal dependencies and knowledge of the repository code from the user end. The code base is also modular and clearly shows the users how to add another dataset and convert it into the common format used by all methods and evaluation scripts, enabling easy extensions to other task settings. With a fresh installation of ConGenBench, one can set up train, test and validation splits for all task settings with fewer than 5 lines of code. This allows researchers to focus on developing the methods and spares time and effort on the testing setup.

---

### Decision · Action_Editor_jeKC · 2024-06-25

**Recommendation:** Reject

**Comment:**

I understand the authors will feel it is unfair to dismiss work based on experimental constraints dictated by budget, and I can sympathize with the limitations this places on research avenues. That said, it is non uncommon in the sciences for such constraints to be a natural part of the life-cycle of projects, and as part of the decision to launch a line of investigation or not (e.g. see particle physics). As such, I agree with the reviewers that the paper is not acceptable until a broader-based comparison of models is made using a sufficient number of annotators. If the authors have the conviction that this method holds water, I hope they can find a method for properly evaluating multiple models against a sufficient annotator base to prove the concept.

**Audience:**

The topic is of interest to a wide community of ML practitioners.

**Claims And Evidence:**

The authors propose a novel method for controllable text generation and showcase its effect on a Mistral model, with a small pool of human evaluators. The method, while interesting, suffers from lack of empirical validation due to the reduced scope of comparison in the experiments.

**Resubmission Of Major Revision:**

The authors may consider submitting a major revision at a later time.